# Absence of Hepatitis E Virus (HEV) in Italian Lagomorph Species Sampled between 2019 and 2021

**DOI:** 10.3390/ani13030545

**Published:** 2023-02-03

**Authors:** Luca De Sabato, Giovanni Ianiro, Virginia Filipello, Sara Arnaboldi, Francesco Righi, Fabio Ostanello, Monica Giammarioli, Antonio Lavazza, Ilaria Di Bartolo

**Affiliations:** 1Department of Food Safety, Nutrition and Veterinary Public Health, Istituto Superiore di Sanità, Viale Regina Elena, 299, 00161 Rome, Italy; 2Department of Food Safety, Istituto Zooprofilattico Sperimentale della Lombardia e dell’Emilia-Romagna “Bruno Ubertini” (IZSLER), Via Antonio Bianchi, 7/9, 25124 Brescia, Italy; 3Department of Veterinary Medical Sciences, University of Bologna, Via Tolara di Sopra, 50, 40064 Ozzano dell’ Emilia, Italy; 4Istituto Zooprofilattico Sperimentale dell’Umbria e delle Marche “Togo Rosati” (IZSUM), Via Salvemini, 1, 06126 Perugia, Italy; 5Department of Animal Health and Welfare, Istituto Zooprofilattico Sperimentale della Lombardia e dell’Emilia-Romagna “Bruno Ubertini” (IZSLER), Via Antonio Bianchi, 7/9, 25124 Brescia, Italy

**Keywords:** hepatitis E virus, rabbit, hare, wild rabbit, zoonosis, HEV, Italy, qRT-PCR, ELISA, wildlife

## Abstract

**Simple Summary:**

Hepatitis E is an emerging zoonosis. In the European Union, human cases are mainly caused by foodborne infections linked to the consumption of raw or undercooked food containing liver from pigs or wild boars infected by the hepatitis E virus (HEV). Some animal species are suspected to be a reservoir of zoonotic HEV genotypes 3 and 4. Among the HEV-3 genotype, viral strains infecting rabbits (HEV-3ra) are strictly related to the zoonotic genotype detected in pigs and humans. However, the transmission of rabbit HEV to humans is still debatable. To better evaluate the risk of transmission of rabbit HEV to humans, it is important to assess its prevalence in lagomorph populations. The aim of this study was to evaluate the presence of HEV in 328 Italian hares and 59 farmed rabbits collected in 3 Italian macro-areas (North, North-Central, and South-Central) between 2019 and 2021. For this purpose, liver samples were tested to detect HEV RNA using broad-range PCR, which was able to detect both HEV-3 and HEV-3ra. Neither HEV RNA nor anti-HEV antibodies were detected. The circulation of HEV-3ra in Italy may be limited to some geographical regions, as confirmed by its absence in our study and by the lack of human cases reported so far.

**Abstract:**

The zoonotic hepatitis E virus genotype 3 (HEV-3) causes most autochthonous human hepatitis E cases in Europe, which are due to the consumption of raw or undercooked food products of animal origin. Pigs and wild boars are considered the main reservoirs of this genotype, while rabbits are the reservoir of a distinct phylogenetic group named HEV-3ra, which is classified within the HEV-3 genotype but in a separate clade. Evidence for the zoonotic potential of HEV-3ra was suggested by its detection in immunocompromised patients in several European countries. HEV-3ra infection was found in farmed and feral rabbit populations worldwide and its circulation was reported in a few European countries, including Italy. Furthermore, Italy is one of the major rabbit meat producers and consumers across Europe, but only a few studies investigated the presence of HEV in this reservoir. The aim of this study was to assess the presence of HEV in 328 Italian hares and 59 farmed rabbits collected in 3 Italian macro-areas (North, North-Central, and South-Central), between 2019 and 2021. For this purpose, liver samples were used to detect HEV RNA using broad-range real-time RT-PCR and nested RT-PCR. Using 28 liver transudates from hares, the ELISA test for anti-HEV IgG detection was also performed. Neither HEV RNA nor anti-HEV antibodies were detected. Further studies will be conducted to assess the HEV presence in Italian lagomorphs to establish the role of this host and the possible risk of transmission for workers with occupational exposure, to pet owners and via food.

## 1. Introduction

Hepatitis E virus (HEV) causes self-limiting, asymptomatic or symptomatic hepatitis that can become chronic in immunocompromised patients. Among the HEV strains belonging to the *Paslahepevirus balayani* species [1], four major genotypes (named HEV-1 to HEV-4) that infect humans are recognized [2]. HEV-1 and HEV-2 are restricted to humans, circulate mostly in low-income countries and are transmitted through the fecal–oral route. HEV-3 and HEV-4 genotypes infect humans and other mammals, and are responsible for zoonotic transmission, causing hepatitis linked to the consumption of raw or undercooked meats, such as pig liver sausages or wild boar meat [3]. HEV-3 and HEV-4 circulate in pigs, which comprise the main reservoir; in wild boars; and, to a lesser extent, in deer [4]. Rabbits are the reservoir of HEV-3ra, which is strictly related to HEV-3 but belongs to a different clade [5,6].

Additional genotypes were described in recent years in wild boars in Japan (HEV-5 and HEV-6), and in camels in the Middle East (HEV-7) and China (HEV-8) [7,8,9]. Among them, only the HEV-7 is zoonotic, where it was also reported in patients with a proven epidemiological link with camel-derived food consumption (raw milk and meat) [7].

The first report of the HEV-3ra strain was in farmed Rex rabbits, which is a breed of European rabbit (*Oryctolagus cuniculus*) in China [10]. Infection in rabbits is apparently asymptomatic, although subclinical hepatitis was observed in experimental infections [11]. Experimental infection of rabbits with HEV-3 genotype strains resulted in the successful detection of HEV in the feces of animals not corresponding to viremia starting from 2 days post-infection. The viral shedding lasted for 5 weeks, leaving open the question regarding the ability of lagomorphs to shed HEV infectious particles for a long time in the environment [12].

Rabbit HEV strains are evolutionarily correlated to HEV-3 genotype strains but form a separated clade from the other HEV-3 strains [8,13], sharing 76–79% nucleotide identities with other HEV genotypes [14]. The zoonotic potential of HEV-3ra was confirmed by successful experimental infection of non-human primates [15] and by detecting the HEV-3ra in acute and chronically infected patients in France [16], Switzerland [17], Spain [18] and in blood donors in Ireland [19].

Several studies reported HEV-3ra worldwide in wild, laboratory and domestic rabbits with variable prevalence: 16.4% in the United States [20], 3.5% in Australia [21], 12.0% from laboratory rabbits [22] and 11.4% from farmed rabbits in China [23], and 6.4% in farmed rabbits from Korea [24]. Overall, the prevalence is variable and significantly lower than that observed in pigs [4].

In Europe, HEV-3ra was also reported in France (7.0% farmed rabbits and 22.9% wild rabbits) [25], in Germany (17.1% wild rabbits, 25.0% feral European rabbits) [26,27], in Poland (14.9% slaughtered farmed rabbits) [5] and in the Netherlands (22.8% farmed, 0.0% wild, 60.0% pet rabbits) [28]. In Spain, wild rabbits (n = 372) and Iberian hares (*Lepus granatensis*) (n = 78) tested negative for HEV [29], while in Portugal, a seroprevalence of 4.1% was reported [30]. In addition, a long-established circulation of rabbit HEV in Europe was confirmed by the detection of an HEV sequence in repository sera collected in 1989 in Germany [31].

In Italy, HEV in rabbits was first reported in a pet house rabbit (*Oryctolagus cuniculus*). The detected strain was closely related to a human HEV-3 sequence via phylogenetic analysis [32]. Furthermore, two HEV-3 and two HEV-3ra strains were detected from wild rabbits (*Oryctolagus cuniculus*) hunted in Toscana region, Central Italy (4/35; 11.4%) [33]. The same study reported a seroprevalence of 42.8% [33], while another study involving animals sampled in the same area reported a seroprevalence of 38.5% [34]. Similarly, a previous study evidenced circulation of HEV in Italian rabbits sampled in several Adriatic regions of Eastern Italy (Marche, Puglia, Emilia-Romagna), with an anti-HEV seroprevalence of 3.4% in 206 farmed rabbits and 6.5% in 122 pet rabbits, but no evidence of HEV RNA in the feces of the investigated animals was obtained [35].

Detection in hares (family *Leporidae*, genus *Lepus*) is less frequent [14] or absent, as reported in Spain regarding 450 livers tested for HEV RNA [29] or in the 47 liver samples from healthy hares hunted in Northern Italy [36]. A small population of brown hares in the Toscana region (Central Italy) also tested negative for the detection of antibodies against HEV [34].

Evidence of a limited HEV-3ra infection in European brown hares (*Lepus europaeus*) was also obtained in Germany. The HEV-3ra RNA was not detected in the first study in any of the 624 hare livers tested [26], but it was revealed in 1 out of 2389 (0.04%) sera from a second study [14]. In the same study, the exposure of hares to HEV was confirmed through the detection of specific antibodies, albeit the detection of seroprevalence (2.2–2.6%) was lower than those observed in wild rabbits (37.3%) from the same country [14,26]. The genome analysis of the HEV strain detected in the serum of the only HEV-positive hare shared 86% nucleotide identity with the HEV-3ra genotype [14].

Italy, after Spain and France, is the third main producer and consumer of rabbit meat in the EU, with 24.5 million rabbits slaughtered in approved rabbit slaughterhouses and 4.5 million slaughtered in backyard farms for direct and local sales. The annual per capita consumption of rabbit meat is 0.5–1 kg. Most rabbits are locally produced in small and large farms across the country [37]. Furthermore, hunting activities are also frequent and hunted animals are mostly locally consumed with limited sanitary inspections. The transmission of HEV-3 from rabbits, as described for pigs and wild boar, may occur either through the consumption of meat from infected rabbits and hares or as a result of occupational exposure in breeders, veterinarians, slaughterers and hunters. Nevertheless, the role of this putative reservoir of HEV is poorly known in Italy.

Therefore, the aim of the present study was to evaluate the role of hares and domestic rabbits as possible reservoirs of HEV.

## 2. Materials and Methods

### 2.1. Sampling

The sampling of lagomorphs included in this study was performed in the framework of the regional passive surveillance plans for wild fauna diseases, which are active in Italy. Liver samples were collected from adult rabbits that died for unknown reasons in cages during farming. Overall, livers from 328 brown hares and 59 farmed rabbits were sampled.

Livers of hares and farmed rabbits were collected from 13 and 6 Italian regions, respectively. Based on the geographical distribution of samples and the density of lagomorphs, three macro-areas were defined as follows: 73 livers from Northern Italy (Piemonte, Lombardia and Veneto regions; area A), 157 from North-Central Italy (Emilia-Romagna and Toscana regions; area B) and 98 from South-Central Italy (Umbria, Marche, Lazio, Abruzzo, Molise, Campania, Basilicata and Calabria regions; area C) (Figure 1). In Italy, the mean density of hares is variable and poorly known. It is estimated that there are up to 10–50 hares/ha, declining from North to South, with a higher density in Emilia-Romagna and a peak in Toscana (25–80 hares/ha) (both regions constituting the whole area B of this study) [38]. The rabbits were housed mainly in small-scale farms, where 49 livers were collected from animals farmed in South-Central Italy regions included in area C (Umbria, Marche, Lazio, Abruzzo) and 10 from area B.

The sample size was calculated to provide a 95% probability of detecting HEV RNA at a 4% design prevalence and considering an infinite (>5000) hare population for each area.

### 2.2. Nucleic Acid Extraction and Evaluation of the RNA Recovery Rate

Total RNA was extracted from 100 mg of tissue, homogenized using the TissueLyser LT (Qiagen, Monza, Italy) and then subjected to RNA extraction using the RNeasy Mini Kit (Qiagen, Monza, Italy), as previously described [39].

To evaluate the rate of recovery of RNA extraction, 75 liver samples, before homogenization, were artificially contaminated with 10 μL of murine norovirus (MuNoV, strain: MNV-IT1 Acc. No. KR349276 [40]) as an extraction process control. Thereafter, the RNA was tested using real-time RT-PCR for the detection of MuNoV using the QuantiFast Pathogen + IC Kits (Qiagen, Monza, Italy), as previously described [39]. The recovery rate was estimated using the comparative cycle threshold (Ct) method [41]. All spiked samples were positive for MuNoV (mean recovery rate of 65.22% ± 0.37).

### 2.3. HEV Real-Time RT-qPCR and Nested RT-PCR

HEV real-time RT-qPCR was performed using the QuantiFast Pathogen + IC Kits (Qiagen), as previously described [42].

The primers and probes used in the real-time qRT–PCR [43] were HEV-F (5’-GGTGGTTTCTGGGGTGAC-3’), HEV-R (5’-AGGGGTTGGTTGGATGAA-3’) and HEV probe (TaqMan HEV probe, 5’-FAM-TGATTCTCAGCCCTTCGC-BGQ1-3’). HEV Real-time RT-qPCR was performed with reverse transcription at 50 °C for 30 min, followed by a denaturation step of 2 min at 95 °C, and 40 cycles at 95 °C for 15 s, 55 °C for 20 s, and 72 °C for 20 s.

Ten microliters of the obtained extracted RNA were reverse-transcribed with HEV-40 Rw primer [44] using the SuperScrip III Reverse Transcriptase kit (Thermo Fisher Scientific, Monza, Italy). The reverse transcription was performed at 50 °C for 30 min and 72 °C for 15 min to inactivate the enzyme. The first round and the nested PCR were performed using the GoTaq G2 Flexi DNA Polymerase (Promega, Milan, Italy) by amplifying a 412 bp fragment within the ORF2 (positions 5953–6363 respect to the E116-YKH98C reference strain, Acc. No. AB369687), as previously described [44]. All reactions were performed including DEPC water, and RNA obtained from an HEV-3-positive liver as negative and positive controls, respectively. Briefly, the first round was performed using a PCR protocol of 2 min at 95 °C, and subsequent 35 cycle at 94 °C for 1 min, at 55 °C for 30 s and 72 °C for 1 min, followed by a final extension at 72 °C for 7 min. Nested PCR was performed at 95 °C for 2 min, followed by 40 cycles at 94 °C for 30 s, 56 °C for 30 s and 72 °C for 45 s.

### 2.4. Antibodies Detection

Transudate from livers (“liver juice”) was recovered from 28 hares. Liver samples were frozen at −80 °C and thawed at room temperature twice. The resulting liquid was collected after a few hours [45] and used to detect the total anti-HEV antibodies with a multispecies ELISA KIT (ID Screen® Hepatitis E Indirect Multi-species ID-Vet, France) following the manufacturer’s instructions.

## 3. Results

A total of 387 liver samples from lagomorphs, namely, 59 farmed rabbits and 328 brown hares, were analyzed to evaluate the presence of HEV RNA. 

The efficient RNA extraction from liver samples was confirmed using a mean recovery rate of >65%. HEV RNA was not detected in the 387 livers analyzed using either real-time RT-qPCR or nested RT-PCR. The HEV prevalence was 0% in all macro-areas. The confidence intervals (95% CIs) calculated using the binomial method based on the beta distribution were as follows: area A, 95% CI: 0.00–4.93; area B, 95% CI: 0.00–2.32; and area C, 95% CI: 0.00–3.69.

The exposure of animals to HEV was also assayed by detecting anti-HEV antibodies in hares from the liver transudates. The results of the ELISA confirmed molecular findings and none of the transudates were positive for antibodies (0/28).

## 4. Discussion

This study was based on the previous evidence of the presence of HEV-3ra in lagomorphs in the Toscana region (Central Italy) [33] and on the widespread circulation of HEV-3 in wild boar in Italy, which may also be present in hares since both are wild and live in the same habitats [33,39,46].

This study involved hares and a small group of farmed rabbits to evaluate the presence of HEV in two populations with possible contacts, which could determine the risk of transmission, as suggested between domestic pigs and wild boars for HEV-3 [4]. However, the circulation of either HEV-3ra or HEV-3 was not confirmed in our study.

The main differences with previous studies were the geographical origin of animals and the species investigated, namely, hares in this study and wild rabbits in previous studies [33,34]. In fact, the detection of both HEV-3 and HEV-3ra was previously reported in Italy only in wild rabbits captured by hunters in a small area of the Toscana region [33]. The circulation of HEV in wild rabbits was also confirmed by the detection of antibodies against HEV in animals captured in the same area (Toscana), where two studies showed seroprevalences of 38.5% (5/13) [34] and 42.8% (15/35) in wild rabbits [33]. Conversely, the same study did not show evidence of antibodies against HEV in the 103 hares investigated, which were captured in the same area as wild rabbits [34].

In this study, we did not investigate wild rabbits; we investigated hares instead, which could account for the differences observed. Furthermore, the hares in our study were collected from the Eastern and Northern parts of Italy, where there is a lower density of hares and wild rabbits compared with the Toscana region [47,48].

We cannot definitively conclude that the virus was not circulating in other Italian regions, as would be suggested by the absence of both HEV RNA and anti-HEV antibodies, but the HEV circulation could likely be very limited in hares and wild rabbits, each representing a possible “spill-over host”. The absence of anti-HEV antibodies in hares in this study could be a bias due to the limited number of liver transudates (n = 28) analyzed, which, as recently published [45], is a good alternative matrix in the absence of sera, to detect antibodies against HEV. However, the result showing that none of the transudates were positive for antibodies is in line with other findings reported in Italy [34,36] and with some studies conducted in the EU. In fact, HEV was not detected in hares in Spain [29] and even in Germany, where HEV was previously detected in hares (0.04%), the percentage was significantly lower than that detected in wild rabbits (2.2%) [14,26].

In our previous study on companion and farmed rabbits that were collected mostly from the same regions involved in this study (the Eastern and Northern parts of the country), the seroprevalence was 3.5% and HEV RNA was also absent [35].

Regarding the HEV RNA detection, the efficient recovery rate of RNA and the use of two molecular broad-range detection methods corroborated our negative results. The real-time RT-qPCR was designed to detect all genotypes, including HEV-3ra [43,49]. Nevertheless, we also tested RNA samples with a broad-range RT-PCR using primer annealing on the consensus sequence within the ORF2 [44], taking into account the possible failure of the real-time RT-qPCR on some HEV-3ra strains [50]. The HEV-3ra strains are variable and the group includes strains with a nucleotide exchange in the overlapping region of open reading frames ORF2/ORF3 of their genomes, possibly interfering with the real-time RT-qPCR reaction [50].

However, the samples were not randomized since hares were collected using passive surveillance of dead animals, some of which may have been affected by an infectious disease of unknown origin. The number of farmed rabbit samples was very low, and we cannot exclude HEV circulation among them. In our previous study on farmed rabbits, HEV RNA was not detected but the detected seroprevalence of 3.4% suggested the circulation of the virus in the population investigated [35]. A low seroprevalence was also observed in other studies in the EU in Poland [5] and the Netherlands [28], suggesting that several factors can contribute to the frequency of infections, such as the herd size, type of farming and age of the rabbits [5,20]. In Poland, a seroprevalence of 6.02% was observed in rabbits housed in small-scale farms and was absent in animals from large-scale farms [5]. Information on the replication of HEV in rabbits was missed, but some studies observed a higher frequency in adult animals at slaughter age, i.e., older than 5 months of age [51,52]. The information on the age of farmed rabbits investigated in this study was missed; indeed, we cannot speculate on this point to justify the negative results observed and should be focused on in future studies.

The absence of antibodies against HEV in this study may suggest that a low circulation or a decreasing prevalence of HEV in rabbits over recent years likely occurred and that hares were a less frequent reservoir than wild rabbits. The latter had different behavior from the hares living in-group with a greater opportunity of transmission via fecal–oral route infection. The source of HEV-3 detected in wild rabbits in the Toscana region [33] could be the infected feces of wild boar. In Italy, several studies revealed that wild boar is a main reservoir of HEV-3 with a high prevalence [36,39,46]. The absence of HEV-3 could be linked to an ecological separation between hares and wild boars, mainly in terms of feeding, which caused a gradual and increasing exclusion of wild boars from rural areas [53]. There are no data on wild rabbits in the regions from where hares of this study were sampled and the seroprevalence of 3.5% reported in farmed and companion rabbits sampled from the same area [35] suggests a limited circulation of the virus. Besides the very low number of rabbit livers tested (n = 59), the limited circulation in farmed rabbits may also be confirmed since neither the previous study [35] nor this study revealed HEV RNA in farmed rabbits.

Despite differences in the HEV prevalence in rabbits among European countries, human infections caused by HEV-3ra were reported in France [16,25], Switzerland [17], Spain [18], Belgium [54] and Ireland [19]. The exposure risk for workers was also demonstrated. In rabbit slaughterhouse workers, a significantly higher seroprevalence and an approximately 6.9-fold increased risk for being seropositive for anti-HEV IgG were observed indeed [55]. Several pieces of evidence support the idea that HEV-3ra may represent another genotype of concern for the population. However, it is not yet clear how rabbit HEV-3ra can contribute to the epidemiology of HEV infection in patients since few epidemiological data were available for the immunocompromised patients that tested positive for HEV-3ra [17,18]. In France, the risk of being infected was higher in subjects who declared that they ate pig, wild game and wild rabbit meat, but no direct evidence of foodborne transmission for HEV-3ra existed [56]. The increased growth of rabbits as companion animals is another potential route of transmission to humans, as may be suggested by the detection of HEV-3 strain in one pet rabbit [32] and the presence of anti-HEV antibodies in pet rabbits [35]. All these findings corroborate the need for investigating the role of lagomorph species as hosts of HEV and as a reservoir in the HEV cycle of transmission to humans.

## 5. Conclusions

Despite the absence of evidence of HEV circulation in Italy in hares and domestic rabbits during the study period, further studies should be conducted to monitor possible fluctuations in the HEV epidemiology and the consequent risk of transmission of HEV-3ra to humans and other wild mammals. The information on HEV circulation in lagomorphs is scarce and would deserve further investigation, including both wild and domestic animals. The possible circulation of zoonotic HEV in these species is a major concern in Italy, which is one of the largest European producers and consumers of rabbit meat, with an extended farming activity across the country. For the mentioned reasons, the surveillance of HEV in farmed and wild animals is crucial to assess the future spreading mechanisms of this zoonotic pathogen from lagomorph to humans. Studies on farmed rabbits will give useful information for the evaluation of the biosecurity measures to be implemented in intensive or backyard farms to maintain the food products of rabbit origin as safe for humans and to limit the possible risk of exposure for farmers or slaughterhouse workers.

## Figures and Tables

**Figure 1 animals-13-00545-f001:**
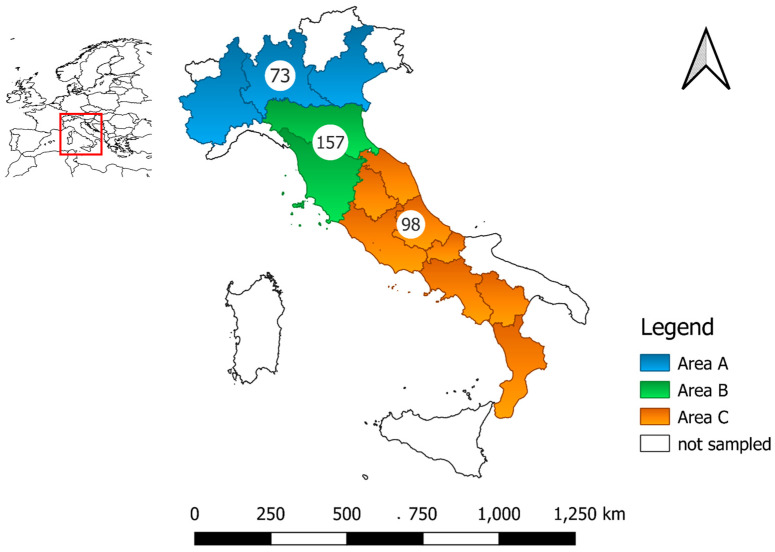
Map of Italy indicating the three areas of sampling and number of livers sampled.

## Data Availability

Not applicable.

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
