# Peer review of "Absence of Hepatitis E Virus (HEV) in Italian Lagomorph Species Sampled between 2019 and 2021"

_animals, 2023, doi:10.3390/ani13030545_

Round 1

Reviewer 1 Report

Authors aims to evaluate the role of hares and domestic rabbits from Italy as possible reservoirs of HEV-3 and HEV-3ra.

Minor comments/suggestions are highlighted in the attached .pdf file.

The manuscript has a high quality, clearly presented. The surveillance of HEV is very important worldwide, in order to decrease the knowledge gap on its transmission and reservoirs considering its zoonotic potential.

Author Response

Authors aims to evaluate the role of hares and domestic rabbits from Italy as possible reservoirs of HEV-3 and HEV-3ra.

Minor comments/suggestions are highlighted in the attached .pdf file.

The manuscript has a high quality, clearly presented. The surveillance of HEV is very important worldwide, in order to decrease the knowledge gap on its transmission and reservoirs considering its zoonotic potential.

REPLY: The Authors thank the Reviewer for the positive comments on the paper. We have accepted and followed all your comments.

why 4%?

REPLY: In the absence of data specifically referred to hares, we considered the lowest prevalence observed in the EU in farmed rabbits that was 7% (France). To be more inclusive we further reduced the value to 4% since in our study we were investigating mainly hares that being wild a prevalence lower than 7% was expected.

Reviewer 2 Report

This article describes serological and molecular methods used for detection of hepatitis E virus in Italian lagomorph species. Since the previous studies conducted in Italy confirmed circulation of HEV in Italian rabbits (based on serological survey and reported case in pet rabbit) this study tried to assess the HEV presence in Italian domestic and wild rabbits sampled in three Italian macro-areas.  However, I do suggest certain things, which need attention, improvement and clarification to support and strengthen the overall impact of the article.

Points for attention:

Title: Authors should think about changing the title.

Simple summary is not needed.

Abstract: Please state the importance of this study. For example, authors can use the fact that it is known that the Italian rabbit meat production and farming sector is larger than in other EU countries. Moreover, since rabbit HEV strains belong to the zoonotic genotype 3, the possible occupational risk of infection in farmers, veterinarians and slaughterhouse workers, but also in rabbit pet owners should be also considered.

Line 42. Maybe it is better to write that the circulation of HEV in Italy may be limited (since the authors stated that HEV RNA wasn’t detected, circulation of either HEV-31 or HEV-3 was not confirmed)

Keywords: To increase article’s searchability it would be good to add more keywords, for example: qRT-PCR, ELISA, farmed rabbits etc. Also, it is better to write wild rabbits than wild animals.

Introduction:

It would be good if the authors would explain the importance of this study, the same was stated in the abstract.  

Materials and Methods:

Figure 1. Map of Italy, indicating three areas of sampling and number of sampled liver samples

Line 125- 2.2. Nucleic Acid Extractions and HEV Real-Time RT-qPCR please rephrase. Did You use more extraction methods?

Results:

Table 1. includes only results for 328 samples, not total of 387 samples. In my opinion Table 1 can be removed because all results are described in the text (Lines 144-146).

Lines 149- 152 should be moved to the chapter 2.3. Antibodies detection

Discussion:

I really don’t understand one thing, first the authors stated in lines 158-159 that the main difference to previous study is the geographical origin, then in lines 165-166 the authors stated that the samples in this and previous study were mainly sampled from eastern and northern parts of the country (samples in both studies are from the same geographical origin). So, what statement is true?

In the line 182 authors stated that the nested RT-PCR was performed but this method wasn’t mentioned earlier, nor it was described in the materials and methods chapter, it is not enough to mention the usage of other methods (methods that are not described in M&M chapter) and only to cite the reference.

Conclusions:

State the possible reason and importance of negative findings.  Needs improvements in addition of significance impact of the study.

Conclusive remarks: I think that the authors need to use wild and farmed/domestic rabbits throughout the manuscript (somewhere they used farmed rabbits and wild rabbits, somewhere brown hares, somewhere hares, somewhere rabbits).  The article needs several improvements in the English Style, as well as formatting to improve the readability of the document.

Author Response

This article describes serological and molecular methods used for detection of hepatitis E virus in Italian lagomorph species. Since the previous studies conducted in Italy confirmed circulation of HEV in Italian rabbits (based on serological survey and reported case in pet rabbit) this study tried to assess the HEV presence in Italian domestic and wild rabbits sampled in three Italian macro-areas.  However, I do suggest certain things, which need attention, improvement and clarification to support and strengthen the overall impact of the article.

REPLY: we acknowledge this reviewer for his precious suggestions. We agreed with your suggestions and the text was modified accordingly.

Points for attention:

Title: Authors should think about changing the title.

REPLY: we totally agree. The word “circulation” was removed from the title.

Simple summary is not needed.

REPLY: the simple summary was included following the author guideline of the journal.

Abstract: Please state the importance of this study. For example, authors can use the fact that it is known that the Italian rabbit meat production and farming sector is larger than in other EU countries. Moreover, since rabbit HEV strains belong to the zoonotic genotype 3, the possible occupational risk of infection in farmers, veterinarians and slaughterhouse workers, but also in rabbit pet owners should be also considered.

Line 42. Maybe it is better to write that the circulation of HEV in Italy may be limited (since the authors stated that HEV RNA wasn’t detected, circulation of either HEV-31 or HEV-3 was not confirmed)

REPLY: the abstract was changed following the reviewer’s suggestions. We clearly highlighted, as suggested, the importance of rabbit meat production and farming in Italy and the subsequent importance to investigate the role of this reservoir. Some sentences were included in the abstract, introduction and used in the discussion to highlight the importance of this investigation.

Keywords: To increase article’s searchability it would be good to add more keywords, for example: qRT-PCR, ELISA, farmed rabbits etc. Also, it is better to write wild rabbits than wild animals.

REPLY: the keywords were improved accordingly.

Introduction:

It would be good if the authors would explain the importance of this study, the same was stated in the abstract. 

REPLY: a sentence was added accordingly at lines 116-121.

Materials and Methods:

Figure 1. Map of Italy, indicating three areas of sampling and number of sampled liver samples

REPLY: the typo was corrected.

Line 125- 2.2. Nucleic Acid Extractions and HEV Real-Time RT-qPCR please rephrase. Did You use more extraction methods?

REPLY: we apologize, modified as suggested. The method was one.

Results:

Table 1. includes only results for 328 samples, not total of 387 samples. In my opinion Table 1 can be removed because all results are described in the text (Lines 144-146).

REPLY: we agree with the reviewer, the table was removed.

Lines 149- 152 should be moved to the chapter 2.3. Antibodies detection

REPLY: moved as suggested.

Discussion:

I really don’t understand one thing, first the authors stated in lines 158-159 that the main difference to previous study is the geographical origin, then in lines 165-166 the authors stated that the samples in this and previous study were mainly sampled from eastern and northern parts of the country (samples in both studies are from the same geographical origin). So, what statement is true?

REPLY: we apologize, the text was not clear. Both sentences were changed to describe clearly the geographical difference of sampling. In this study both hares and wild rabbits were from East and North of Italy, in a previous study conducted by us on rabbits the animals were from East and North of Italy. Conversely, the sole study detecting HEV-RNA in wild rabbits was conducted on animals captured in Toscana region. From the same area, hares were also captured and investigated, resulting negative for HEV-RNA.

In the line 182 authors stated that the nested RT-PCR was performed but this method wasn’t mentioned earlier, nor it was described in the materials and methods chapter, it is not enough to mention the usage of other methods (methods that are not described in M&M chapter) and only to cite the reference.

REPLY: we apologize, the rev is right. The nested RT-PCR protocol was added in the M&M section. Furthermore, details on the protocols followed were added.

Conclusions:

State the possible reason and importance of negative findings.  Needs improvements in addition of significance impact of the study.

REPLY: the conclusions were improved accordingly.

Conclusive remarks: I think that the authors need to use wild and farmed/domestic rabbits throughout the manuscript (somewhere they used farmed rabbits and wild rabbits, somewhere brown hares, somewhere hares, somewhere rabbits). The article needs several improvements in the English Style, as well as formatting to improve the readability of the document.

REPLY: we agree, the terms sometime were confused. The study was performed on two different lagomorph species, wild rabbits and brown hares. For rabbits, the words “wild” and “farmed” were added throughout the text. The nomenclature “hares” cannot be changed, since it represents an important difference to previous studies investigating only the wild rabbits.

The paper was revised by English mother tongue.
